# Pediatric Primary Hepatic Tumors: Diagnostic Considerations

**DOI:** 10.3390/diagnostics11020333

**Published:** 2021-02-18

**Authors:** Bryony Lucas, Sanjita Ravishankar, Irina Pateva

**Affiliations:** 1Rainbow Babies and Children’s Hospital—Department of Pediatrics, University Hospitals Cleveland Medical Center, Cleveland, OH 44106, USA; 2Rainbow Babies and Children’s Hospital—Department of Pathology, University Hospitals Cleveland Medical Center, Cleveland, OH 44106, USA; 3Rainbow Babies and Children’s Hospital—Department of Pediatric Hematology and Oncology, University Hospitals Cleveland Medical Center, Cleveland, OH 44106, USA

**Keywords:** pediatric, liver tumor, hepatocellular carcinoma, hepatoblastoma, malignant rhabdoid tumor, angiosarcoma, undifferentiated embryonal sarcoma, infantile hepatic hemangioma, hepatic adenoma, mesenchymal hamartoma, focal nodular hyperplasia

## Abstract

The liver is the third most common site of abdominal tumors in children. This review article aims to summarize current evidence surrounding identification and diagnosis of primary hepatic tumors in the pediatric population based upon clinical presentation, epidemiology, and risk factors as well as classical imaging, histopathological, and molecular diagnostic findings. Readers will be able to recognize the features and distinguish between benign and malignant hepatic tumors within different age groups.

## 1. Introduction

The liver accounts for 5–6% of all intra-abdominal masses detected in children, with renal being the most common [1]. One-third of these masses are considered to be benign, whereas two-thirds are malignant [2]. The majority of these masses present in a similar way, with progressive abdominal distension, a palpable abdominal mass, abdominal pain, and hepatomegaly. In pediatrics, the differential for a benign hepatic mass includes infantile hepatic hemangioma, hepatic adenoma, mesenchymal hamartoma, and focal nodular hyperplasia. Malignant etiologies include hepatoblastoma, hepatocellular carcinoma, malignant rhabdoid tumors, undifferentiated embryonal sarcoma, and angiosarcoma. Within the neonatal and early childhood period, hemangioma and hepatoblastoma are the most common benign and malignant primary hepatic tumors, respectively. In school-aged children and adolescents, adenomas and hepatocellular carcinoma are the most common. This review encompasses epidemiology, risk factors, clinical presentation, laboratory findings, diagnostic imaging, immunohistochemistry, and histopathologic findings of the above tumors, as organized by age and benign versus malignant classification.

## 2. Benign Tumors in Infants and Early Childhood

### 2.1. Hepatic Hemangiomas

Hepatic hemangiomas can be divided into congenital and infantile forms. Congenital hepatic hemangiomas (CHH) develop prenatally, are fully grown at birth, and are much more rare, comprising only 3% of all infantile hemangiomas [3]. They may demonstrate three distinct patterns of progression: those which rapidly involute in infancy, known as “rapidly involuting congenital hemangioma” (RICH); those that do not involute, known as “non-involuting congenital hemangioma” (NICH); or those that partially involute, known as “partially involuting congenital hemangioma” (PICH). 

Infantile hepatic hemangiomas (IHH) are the most common benign vascular tumors in infancy, with a prevalence of 4.5% in term neonates, most commonly in white females [3]. They can be classified into three distinct subtypes: focal hepatic hemangiomas (FHH), multifocal hepatic hemangiomas (MHH), and diffuse hepatic hemangiomas (DHH) [4].

FHH may also develop prenatally and be present at birth. In fact, many authors consider FHH to be the same clinical entity as rapidly involuting congenital hemangioma (RICH), as they both stain negative for GLUT-1, unlike other forms of IHH [3]. 

#### 2.1.1. Clinical Presentation and Laboratory Findings

The clinical course and associated laboratory findings of each infantile subtype is summarized in Table 1. CHH are fully grown at birth and as described above, are named for how quickly they involute. Congenital hepatic hemangiomas have overlapping clinical features with FHH present at birth.

All subtypes of IHH may present asymptomatically or with abdominal distension and hepatomegaly. CHH and FHH are unique in that they develop prenatally and may cause fetal cardiomegaly with subsequent fetal cardiac failure, hydrops, and cardiac insufficiency. As such, fetal echocardiography is recommended throughout pregnancy if CHH/FHH are suspected. Anemia and thrombocytopenia may occur with CHH/FHH but are typically self-limited [5]. 

Alpha feto-protein (AFP) is a protein produced by the fetal liver and yolk sack and is initially elevated at birth, then declines to adult values during the first several months of life. As such, AFP will initially be high in the setting of CHH/FHH; however, it should downtrend according to typical physiologic parameters. If AFP is persistently high or rising, an alternative diagnosis such as hepatoblastoma or others should be considered. 

MHH and DHH can be viewed as a spectrum of disease, with progression from the former to the latter. Most MHH are asymptomatic at presentation and may be identified on routine screening ultrasounds in the setting of cutaneous hemangiomas. If portovenous or arteriovenous shunting is present within MHH, this can lead to high output cardiac heart failure. MHH may progress to a diffuse pattern of involvement (DHH), representing near complete displacement of liver parenchyma, and will frequently manifest with a high morbidity and mortality secondary to massive hepatomegaly, with hepatic failure, compression of surrounding organs and vasculature, and/or abdominal compartment syndrome with subsequent multisystem organ failure [4,5,6]. In addition, due to the enormous tumor burden, patients will experience profound consumptive hypothyroidism secondary to overproduction of type 3 iodothyronine deiodinase, which deactivates thyroid hormones [7]. 

#### 2.1.2. Associated Syndromes and Risk Factors

Notably, FHH has no association with infantile cutaneous hemangiomas, as opposed to MHH and DHH, where screening with abdominal ultrasonography is recommended for infants with more than five cutaneous hemangiomas [8]. Premature infants with low birth weight and a positive family history are at increased risk for development of IHH [4]. Rialon et al. examined risk factors for morbidity and mortality based upon 123 patients with MHH or DHH. The overall reported mortality rate was 16%, with a rate of 38% within patients with the diffuse subtype and 9% amongst patients with multifocal. Patients with congestive heart failure, low output from hypothyroidism, and high output from shunting were present within 48% who died, as opposed to 9% who did not. In addition, abdominal compartment syndrome in patients with diffuse hepatic hemangiomas was a prognostic indicator for death [9]. 

#### 2.1.3. Diagnostic Imaging

For a suspected hepatic hemangioma present at birth, the main differential is between CHH (RICH/NICH) and IH (infantile hemangioma/focal). Imaging characteristics are particularly helpful with this, as is serial imaging to assess for involution. As discussed above, RICH and focal IH are considered by many to be the same clinical entity and as such have similar imaging characteristics.

Ultrasound can be diagnostic of congenital hepatic hemangiomas. They typically have high flow and arterial feeding vessels and may demonstrate direct shunts to the hepatic veins. If hemorrhage and thrombosis occur secondary to the shift from fetal circulation, a central hypodensity may be visualized. It is recommended that an echocardiogram also be performed if shunting is present on ultrasound or if the patient presents with any signs/symptoms of congestive heart failure [5]. On MRI and CT, this lesion appears as a solitary, spherical tumor with robust and rapid enhancement (Figure 1). Calcification and central cystic changes are rarely seen with IH but are present in RICH and NICH [3,6]. In comparison to RICH, NICH is more likely to demonstrate intratumoral shunting on ultrasound and display prominent drainage veins on MRI. NICH rarely displays intratumoral flow voids in comparison to RICH and focal IH [10].

If RICH and NICH are difficult to distinguish based upon initial imaging, serial follow-up imaging will aid in diagnosis, with a subsequent decrease in the size of the lesion suggestive of RICH as opposed to NICH [4]. It is recommended that CHH be monitored with serial ultrasounds until stable size and vascularity is demonstrated on two consecutive scans [11].

Multifocal hepatic hemangiomas appear as multiple, discrete masses that are hypoechoic or have mixed echogenicity on ultrasound [5]. On MRI, they present as homogenously enhancing spherical tumors that are hypointense on T1 sequencing and hyperintense on T2 sequencing in comparison to the normal liver. On CT imaging, they are hypodense lesions that have uniform or centripetal enhancement [12] (Figure 2). 

Diffuse hepatic hemangiomas present with innumerable centripetally enhancing lesions that almost completely replace the normal hepatic parenchyma. These lesions are hypointense on T1-weighted imaging and have a prominent flow void on T2 imaging [13].

Contrast-enhanced ultrasound (CEUS) is a new imaging modality without exposure to radiation or need for sedation. On CEUS, IHH have a characteristic rapid filling phase of the hemangioma, which is typically completed at the end of the arterial phase or by the beginning of the venous phase. It classically will appear iso-enhanced or mildly hyper-enhanced in comparison to normal surrounding liver parenchyma [14]. 

#### 2.1.4. Histopathology and Immunohistochemistry

The largest histologic distinction between these tumor types is based upon immunohistochemical staining for GLUT-1. Congenital hepatic hemangiomas and focal hepatic hemangiomas are negative for GLUT-1, whereas multifocal and diffuse hepatic hemangiomas are positive [11]. 

NICH is composed of large lobules of small vessels with intervening fibrosis. The basement membrane is thinner than seen in RICH. RICH typically has larger vessels and lobules of varying sizes. 

Grossly, FHH is a large tumor characterized by central necrosis, hemorrhage, and/or fibrosis. MHH are small tumors with no central necrosis, and DHH characteristically replace the normal liver parenchyma. IHH are similar histologically to CHH, with a fibrous stroma interposed by thin vascular channels lined with endothelial cells [15] (Figure 3).

### 2.2. Mesenchymal Hamartoma

Mesenchymal hamartoma (MH) is a benign tumor that was first described by Edmondson in 1956 [16]. The majority of cases present prior to the age of two, with a slight male predominance [17,18]. This is the second most common benign hepatic tumor presenting in infancy and young children, after infantile hemangioma [19]. Although the pathogenesis is not clear, the most widely accepted theory is that MH arises late in embryogenesis from the aberrant development of mesenchyme in the portal tracts [18,19,20]. Rarely, MH can undergo malignant transformation into undifferentiated embryonal sarcoma [17]. 

#### 2.2.1. Clinical Presentation and Laboratory Findings

The clinical presentation of MH differs based upon age of the patient as well as tumor size. Approximately 15% to 20% of MH present in the neonatal period and may be identified prenatally on ultrasound [21,22]. Although pathologically benign, perinatal complications such as fetal hydrops, maternal toxemia, preterm labor, and fetal demise may occur [23]. Neonates tend to have a more acute clinical presentation due to rapid growth after birth, secondary to accumulation of fluid within the cystic components of the mass. They may present with massive abdominal distension, potentially leading to respiratory distress or apnea from diaphragmatic compression or compression of surrounding structures [21,22,24]. AFP may or may not be elevated above the higher physiologic baseline in this younger age group, although it should downtrend on serial measurements, unlike malignant tumors [24]. In toddlers, small lesions are generally asymptomatic and often diagnosed incidentally on imaging studies but may present with nonspecific symptoms such as abdominal pain, fatigue, and fever. More commonly, the diagnosis is delayed until the tumor is larger enough to cause mass effect on surrounding structures and more severe symptoms [19]. AFP may be normal or elevated in this older age group due to the presence of normal hepatocytes within the tumor and may mimic the presentation of hepatoblastoma [18]. GGT may be mildly elevated as well [17]. 

#### 2.2.2. Associated Syndromes and Risk Factors

Children with Beckwith–Wiedemann Syndrome (BWS) are at a higher risk of developing several types of tumors, including mesenchymal hamartomas [25,26,27]. Of note, the serum AFP of children with BWS may be elevated at baseline compared to healthy children. Whilst hepatoblastoma is the most concerning and likely diagnosis within the differential for children with BWS presenting with a liver mass and an elevated AFP, mesenchymal hamartoma should remain under consideration to avoid an incorrect diagnosis and treatment course [25]. Appelaniz-Ruiz et al. reported two pediatric cases of mesenchymal hamartoma of the liver associated with germline DICER1 pathogenic variants. DICER1 syndrome is an inherited tumor predisposition syndrome that is commonly associated with pediatric pleuropulmonary blastoma, cystic nephroma, and multinodular goiters. They propose that MH represents a new phenotype of DICER1 syndrome [28]. 

#### 2.2.3. Diagnostic Imaging

In MH, cystic or solid components may predominate. In some cases, cysts may be very small and appear as solid components on ultrasound. Typically, solid tumors are more common at younger ages and are smaller overall [24]. Most frequently, ultrasound will demonstrate a large multi-loculated cystic structure with varying degrees of solid echogenic tissue and internal septations [17,18,29]. A “sieve-like” appearance of the solid components is noted on both ultrasound and MRI, which is consistent with the “swiss cheese appearance” reported on gross pathological examination [17,29]. The tumor is well-circumscribed but does not typically have a true capsule. Calcification and/or hemorrhage within the tumor is rare, and most tumors appear hypodense and hypovascular. CT can assist with delineating the origin of the mass and may reveal it to be intrahepatic, extrahepatic, or to have an exophytic or pedunculated pattern of growth [17]. MRI can also help to establish hepatic origin and demarcate solid from cystic components, septations, and the tumor’s relationship to normal surrounding structures and vasculature. T1- and T2-weighted sequences will reveal hypointensity of solid components compared to adjacent liver due to fibrosis. The cystic components will have high signal intensity on T2-weighted images but have variable T1 signal intensity based upon the protein concentration in the fluid [30]. 

#### 2.2.4. Histopathology

Grossly, cut sections will reveal multiple variably sized cysts with clear or pale yellow serous fluid and or mucoid material. Histologically, they are made up of both epithelial and mesenchymal components. Classically, there will be clusters of normal hepatocytes and architecturally normal bile ducts within a primitive matrix of loose mesenchyme [17,18] (Figure 4). Immunohistochemistry is generally not required for the diagnosis. Recurrent chromosomal rearrangements involving 19q13 (mesenchymal harmartoma of the liver breakpoint 1 (MHLB1)) have been noted [30,31].

## 3. Malignant Tumors in Infants and Early Childhood

### 3.1. Hepatoblastoma

Hepatoblastoma (HB) is a tumor of embryonal origin and is the most common primary liver malignancy of children. The incidence is estimated to be 0.5–1.5 cases per million children between birth to age fourteen, most commonly presenting below the age of four [32]. Congenital hepatoblastomas are diagnosed in utero or within the first 28 days of life and comprise less than 10% of all pediatric hepatoblastomas [33,34]. 

#### 3.1.1. Clinical Presentation and Laboratory Findings

Clinically, neonatal HB tends to present with acute onset of increased abdominal distension and a palpable abdominal mass with resultant respiratory distress and clinical decompensation within the first days to weeks of life [35,36,37]. Delivery of these infants has been reported to be complicated by tumor rupture leading to perinatal hemorrhage and subsequent hemorrhagic shock [34,36]. Therefore, it is recommended that these infants be delivered by cesarean section if HBL is diagnosed antenatally [37]. HB presenting later in childhood presents similarly to other hepatic tumors, with abdominal distension or a palpable abdominal mass often associated with nonspecific symptoms such as anorexia, pain, fatigue, and weight loss [38]. Of note, it has been reported that fractures occur in approximately 15% of children with newly diagnosed HB, most commonly in the ribs and spine, and these children may present with irritability or bone pain [39]. In addition, there are case reports of isosexual precocity due to virilization thought to be secondary to beta human chorionic gonadotropin hormone (β-hCG) secretion by HB; these cases typically present in males under the age of 3 [40]. 

AFP is commonly used as a tumor marker for screening and diagnosis of HB, as it is elevated in approximately 90% of patients; however, this screening marker is neither sensitive nor specific for HB and is commonly elevated in other malignant liver tumors. In small cell type HB, AFP may not rise due to decreased differentiation of cells, but in more advanced and differentiated types of HB such as fetal HB, AFP is usually elevated [35,37]. Patients can also present with marked thrombocytosis, thought to be secondary to thrombopoietin production in tumor tissues [41,42]. 

#### 3.1.2. Associated Syndromes and Risk Factors

HB is associated with Beckwith–Wiedemann Syndrome, trisomy 18, hemihypertrophy, and intestinal polyposis [38]. It is well documented that low birth weight, particularly less than 1500 g, and prematurity are risk factors for the development of HB later in life. In a recent review, it was noted that HB cases were more likely to be born at a younger gestational age, have lower birth weight and length, and have longer neonatal intensive care unit stays than matched controls [43]. Current research in this area is focused on identifying the etiology of this relationship, which appears to be different than normal birth weight children. It is postulated that common NICU exposures that increase oxidative damage (radiation, total parenteral nutrition, antibiotics, oxygen therapy) may be more detrimental to low birth weight children, increasing their likelihood of development of HB [43,44]. Children with lower birth weights tend to be diagnosed later than children with normal or high birth weights [44]. Parental tobacco use has been reported as a risk factor; however, no clear consensus has been reached, and the evidence is controversial [45]. A study by the Children’s Oncology Group examined maternal pregnancy events and exposures in relationship to the risk of hepatoblastoma and found that there was an association with maternal weight gain early in pregnancy independent of the index child’s weight. They hypothesize that this may be related to maternal weight gain specifically during fetal formation of the liver bud at weeks 9 to 10 gestation [46]. Further research is needed in this area to elucidate whether there is a single causative exposure versus multiple neonatal exposures. Similarly, it has yet to be determined whether low birth weight hepatoblastoma cases have an underlying genetic susceptibility. 

#### 3.1.3. Diagnostic Imaging

Congenital HB may be identified prenatally by maternal ultrasounds in the early third trimester and typically present as a single, solid, and well-circumscribed echogenic lesion [47]. Pregnancies are complicated by polyhydramnios secondary to gastrointestinal compression from the mass with resultant fetal hydrops. Fetal MRI may be a useful adjunct in guiding decisions regarding delivery method and immediate perinatal care. CT scans are commonly performed on patients with suspected abdominal masses and can play a crucial role in narrowing the differential diagnosis. The majority of congenital HBs develop within the right lobe of the liver, likely secondary to the anatomy of fetal circulation. Calcification is rare within HB but, if present, is likely to represent an epithelial subtype. Imaging of congenital HB tends to demonstrate a pattern of progressive expansion and fusion filling but significant heterogeneous enhancement on contrast-enhanced CT. This imaging differs from HB in older children, which tends to enhance less than surrounding liver tissue [33] (Figure 5).

#### 3.1.4. Histopathology and Immunohistochemistry

HB can be subcategorized based on their histopathology, summarized in Table 2. The two main types are epithelial and mesenchymal. Epithelial has four main subtypes: fetal, embryonic, small cell, and macrotrabecular. Fetal hepatoblastoma is composed of cells that resemble fetal hepatoblasts during embryonal development. Well-differentiated or pure fetal HB has low mitotic activity (less than two mitotic figures per 10 high power fields) as opposed to crowded HB, which is more mitotically active (greater than two mitotic figures per 10 high power fields). Of note, diagnosis of well-differentiated fetal HB requires analysis of the complete resection specimen and cannot be made on biopsy specimens nor after chemotherapy. The embryonal pattern is the most common subtype and resembles the normal liver histology seen at 6 to 8 weeks’ gestation (Figure 6). Small cell type is associated with an aggressive clinical course, poor survival, and accounts for approximately 5% of all HB cases. Macrotrabecular HB accounts for approximately 5% of HB cases. It demonstrates a growth pattern of cell plates more than twenty cells thick, consisting of fetal, embryonal, or pleomorphic cells, and is similar in appearance to hepatocellular carcinoma [38]. 

Immunohistochemistry can be useful in clarifying histologic subtype in difficult or post-therapy cases. The most common stains are alpha feto-protein, glypican 3, β-catenin, glutamine synthetase, and INI-1. Glypican-3 demonstrates a fine granular cytoplasmic staining pattern unique to the well-differentiated fetal pattern of HB. Β-catenin is a marker of the activated canonical Wnt pathways (thought to be a key driver in the development of HB) and is helpful as only neoplastic tissue demonstrates diffuse cytoplasmic expression without nuclear staining, allowing neoplastic tissues to be differentiated from normal hepatocytes and biliary epithelial cells, which show only membranous staining. Glutamine synthetase, another marker for the activated Wnt pathways, is expressed with high intensity in fetal hepatoblastoma. Pure small cell HB retains INI-1 expression, which allows for differentiation from malignant rhabdoid tumor, which is very similar histologically [38].

### 3.2. Malignant Rhabdoid Tumors

Malignant rhabdoid tumors (MRT) were first described in the kidney in 1982 by Beckwith and Palmer as a variant of Wilms’ tumor [48] and were discovered in the liver in 1982 [49]. Extrarenal MRT are much more rare, with the liver being the fourth most common site [50]. They are most commonly diagnosed in infants and toddlers with the incidence estimated to be 0.6 per 1 million people, with a median age of 11–18 months for the extrarenal subtypes [51]. 

#### 3.2.1. Clinical Presentation and Laboratory Studies

Hepatic MRT frequently presents with fever, abdominal distension, abdominal pain, decreased oral intake, and emesis associated with a right upper quadrant mass or hepatomegaly [50,51,52,53,54,55,56,57,58]. Patients may present with leukocytosis, anemia, transaminitis, elevated LDH with normal to mildly elevated AFP. In comparison to hepatoblastoma, the most common liver tumor in this age group, patients tend to be younger at diagnosis and more frequently present with spontaneous tumor rupture and most notably have a normal or only mildly elevated AFP [59]. 

#### 3.2.2. Diagnostic Imaging

Initial imaging with ultrasound most commonly reveals a heterogenous solid mass, which may have cystic components [51,52,54,55,56,60,61]. CT and MRI imaging typically reveal hypodense areas on contrast-enhanced CT and hyperintense areas on MRI T2-weighted images [50,55,57]. In addition, there may be areas of necrosis and fluid levels suggestive of hemorrhage [51,57]. Calcifications are less common than in hepatoblastoma. 

#### 3.2.3. Histopathology

Classically, the histologic identification of MRT is dependent on the presence of rhabdoid cells: large polygonal cells with eccentric vesicular nuclei, prominent nucleoli, and abundant eosinophilic cytoplasm (Figure 7). However, the presence of rhabdoid morphology may be a minor component or may be absent, making immunohistochemical staining for INI-1 paramount in ensuring accurate diagnosis of MRT. INI-1 is a tumor suppressor gene involved in the SWI/SNF chromatin remodeling complex; almost all reported cases of hepatic MRT to date have identified loss of nuclear INI-1 protein expression, making this a key factor in differentiating it from other hepatic malignancies. 

### 3.3. Angiosarcoma

Pediatric hepatic angiosarcoma is an extremely rare malignant vascular tumor that may occur at any time during the neonatal period up until adolescence, accounting for 0.3% to 2.5% of liver tumors in children [62,63,64]. There is a reported female predominance within the cases, and the median age is reported to be 40 months [62,65,66]. We were able to find 72 published cases of pediatric hepatic angiosarcoma since 1944 (Appendix A) [62,63,64,65,66,67,68,69,70,71,72,73,74,75,76,77,78,79,80,81,82,83,84,85,86,87].

#### 3.3.1. Clinical Presentation and Laboratory Findings

In the neonatal and infantile period, these tumors present with rapid onset abdominal distension secondary to hepatomegaly, which may lead to abdominal compartment syndrome [63,79]. McClean et al. reported a case of a six-week-old infant with hepatic angiosarcoma who presented with findings very similar to those of multifocal and diffuse infantile hemangiomas; however, the child lacked the characteristic findings of consumptive hypothyroidism and high-output cardiac failure. This patient instead presented with elevated β-hCG and hyperthyroidism [63]. In childhood, patients present with an abdominal mass that may be associated with abdominal pain, emesis, jaundice, or respiratory distress [65]. Laboratory evaluation may be notable for consumptive coagulopathy and anemia. AFP is typically normal, as are other common tumor markers, although elevated β-hCG has been reported [62,63,66,80,88]. Pulmonary metastasis is common [63].

#### 3.3.2. Associated Syndromes and Risk Factors

Environmental exposures, such as vinyl chloride, causing hepatic angiosarcoma are frequently reported in adults [89]; however, we found only one reported case of hepatic angiosarcoma secondary to arsenic exposure reported in a 20-month-old female, described by Falk et al. in 1981 [70]. There have been cases reported of hepatic angiosarcoma developing in patients with multiple cutaneous infantile hemangiomas, as well as later in childhood with patients who have a history of infantile hepatic hemangiomas [62,63,79,80,83]. Jeng et al. identified a de novo KRAS L19F mutation that they proposed was responsible for the malignant transformation of an infantile hemangioma [83]. This mutation has also been described in hepatic angiosarcoma secondary to vinyl chloride exposure in adults [90]. In 2014, Olson et al. published the first case report of a 17-year-old female who developed angiosarcoma in the setting of dyskeratosis congenita, a multisystemic disease caused by genetic mutations resulting in defective telomere maintenance [84]. 

#### 3.3.3. Diagnostic Imaging

To date, there have been no classic imaging findings reported that could allow for pre-biopsy diagnosis of hepatic angiosarcoma in pediatrics. In case reports, angiosarcoma has imaging findings typical of many other vascular lesions of the liver. Ultrasound may reveal a mixture of hypoechoic and hyperechoic regions reflective of areas of necrosis and hemorrhage [88]. Large feeding vessels are less commonly identified in angiosarcoma than in infantile hepatic hemangiomas [63]. In addition, angiosarcomas have been reported to behave slightly differently than hemangiomas on arterial phase enhancement after contrast on CT and/or MRI scans. Angiosarcomas tend to exhibit irregular central enhancement in comparison to the peripheral centripetal enhancement typically seen with infantile hepatic hemangiomas [80]. 

#### 3.3.4. Histopathology

Definitive diagnosis of hepatic angiosarcoma is made based on histopathology. It is imperative that histopathologic examination is performed on an adequate sample of tissue that is representative of the tumor as a whole, preferably obtained during resection, as malignancy may be present focally in the background of a benign hemangioma [62,88]. Histologic examination reveals a poorly differentiated and pleomorphic tumor with marked cellular atypia. Cells are highly atypical spindle-like cells with abundant mitoses and may be arranged in a whorl pattern or in bundles with slit-like vascular spaces [63,79,88]. Immunohistochemistry is positive for endothelial cell markers CD31 and CD34 and may be positive for podoplanin, a lymphatic marker [63,64]. 

## 4. Benign Tumors in School-Aged Children and Adolescents

### 4.1. Hepatocellular Adenomas

Hepatocellular adenomas (HCA) are rare benign tumors arising from hepatocytes. HCA comprise less than 5% of all pediatric liver tumors and typically occur in adolescence with a mean age of diagnosis around 14 years old [91,92]. There are, however, reports of HCA being diagnosed prenatally and in infancy [93]. There is a female predominance that is well-reported in adult cases; however, it appears that there may be a more even gender distribution within the pediatric population based upon published case series. HCA are classified into four subtypes based upon their genetic and phenotypic characteristics: hepatocyte nuclear factor 1α (HNF1α)-mutated HCA (H-HCA), β-catenin-mutated HCA (β-HCA), inflammatory HCA (I-HCA), and unclassified HCA. These are well-described in the adult literature, but there is a paucity of literature in the pediatric population [94]. 

#### 4.1.1. Clinical Presentation

Clinically, most HCA are discovered incidentally but, if symptomatic, present with abdominal pain as the chief complaint. The major clinical complication associated with HCA is rupture and subsequent intratumoral or intraperitoneal hemorrhage, described in approximately 25% of cases across all age groups and most commonly associated with the inflammatory subtype [95]. In addition, there is a risk of malignant transformation to hepatocellular carcinoma correlated with increased tumor size (greater than five centimeters), male gender, and associated with β-catenin activation. Malignant transformation most commonly occurs in adulthood, with an estimated occurrence of 4% across all ages; therefore, case reports in children are extremely rare. AFP is typically normal but will start to elevate with malignant transformation [96]. In the setting of pediatric nonalcoholic steatohepatitis (NASH), β-catenin activation and IL-6 pathways have been implicated in malignant transformation [97].

#### 4.1.2. Associated Syndromes and Risk Factors

In pediatrics, HCA may occur spontaneously; however, they most commonly occur in association with a known risk factor or predisposing condition summarized in Appendix A [98]. 

The association between estrogen exposure and development of HCA is well established, but the pathogenesis is poorly understood. HCA are more common in women and obese men, secondary to peripheral production of estrogen from adipose tissue, and are associated with high potency combined estrogen/progestin oral contraceptives [99,100]. Norethindrone and noresthisterone, both synthetic progesterones, have also recently been implicated in HCA development, likely secondary to peripheral conversion to ethinyl estradiol [99,101]. Crosnier et al. reported a case series of four young women (aged 14–24) with recessive inherited platelet disorders who were receiving continuous noresthisterone treatment to induce amenorrhea. All four of these patients developed HCA, two of whom presented with life-threatening hemorrhage secondary to ruptured HCA. All patients had spontaneous regression of HCA after cessation of noresthisterone [101]. Androgens have also been associated with HCA and are commonly used in the treatment of patients with endocrine abnormalities, aplastic anemias, hereditary angioedema, muscle mass development, and in transgender individuals [102]. Endogenous androgen production causing HCA is much less common but has been reported in the setting of an adolescent with polycystic ovarian syndrome and type 2 diabetes mellitus [103]. 

HCA is also associated with glycogen storage diseases. Glycogen storage disease type 1 (GSD1) is an autosomal recessive disorder of metabolism-causing defects in endogenous glucose synthesis. Type 1a (GSD1a) is characterized by deficiency in glucose-6-phosphatase catalytic activity leading to impaired glycogenolysis and gluconeogenesis. Glucose-6-phospahte is shunted into other metabolic pathways, causing hypertriglyceridemia, hyperlactatemia, and hyperuricemia. The current treatment involves dietary therapy with uncooked cornstarch [104]. Hepatic adenomas, with the potential malignant transformation or intratumoral hemorrhage, are considered to be a major cause of morbidity and mortality within this population because of their increased frequency. HCA typically develop during or after puberty in the second and third decade of life, with a reported incidence of approximately 75–80% by adulthood with a 1:1 female-to-male ratio. The pathophysiology of development has yet to be clearly elucidated and is likely a combination of environmental and genetic factors. Wang et al. reported an increase in adenoma development with patients who had higher mean triglyceride concentrations; however, prior studies had not found metabolic control to be a significant contributor. It has been reported that chromosomal aberrations involving a simultaneous gain of chromosome 6p and loss of chromosome 6q are the most common genetic anomalies seen in HCA developing in the setting of GSD1a [105]. In addition, there is a high frequency of β-catenin activation in GSD1a patients who develop HCA, leading to an elevated risk of malignant transformation to hepatocellular carcinoma (HCC) within this population [106].

Congenital extrahepatic portosystemic shunts are a predisposing risk factor for development of HCA. Multiple theories surrounding pathogenesis of HCA in this context have been proposed. It is theorized that development of HCA is related to the anomalous blood supply characterized by excessive arterialization with resultant increased oxygen delivery to hepatic tissue. In addition, alterations in blood supply to the liver with diversion of splanchnic blood flow may result in abnormal composition of hepatotrophic substances such as insulin and estrogen, which have been clearly linked to development of HCA [107]. Support for these theories is demonstrated by full or partial regression of HCA after shunt closure and normalization of blood flow [108]. 

Adult survivors of childhood and young adult cancer are at risk for development of hepatic adenomas. A case series of twelve patients found that female gender, history of stem cell transplant, hormone replacement therapy, and total body irradiation were associated with development of HCA. These tended to be large, multiple, and primarily of the inflammatory subtype [109].

#### 4.1.3. Diagnostic Imaging

Ultrasound is typically the first line imaging tool utilized in children with a suspected liver mass and as a screening tool in patients with known risk factors; however, it can frequently miss small isoechoic nodules or those within a background of steatosis. HCA are classically described as heterogenous, well-delineated solid masses. On MRI, HCA are commonly heterogeneous on T1- and T2-weighted images, with a high signal on T2-weighted images. Early enhancement is typically seen with IV contrast administration, and a pseudo-capsule can be observed on delayed acquisition images. Early washout after early enhancement in the arterial phase with IV contrast is suggestive of malignant transformation to HCC [91]. On contrast-enhanced ultrasound (CEUS), HCA demonstrate isoenhancement on the early and late portal venous phase [91,92]. Of note, patients with portosystemic shunts will lack portal vascularization, and they therefore have an absence of the early arterial enhancement classically seen in HCA. In children with GSD, annual US is recommended to screen for HCA. 

#### 4.1.4. Histopathology

Grossly, HCA are soft, well-demarcated tumors with minimal to no fibrous capsule present. HCA are composed of normal-sized hepatocytes arranged in mildly thickened or irregular liver cell plates with a parenchyma supplied by numerous arteries unaccompanied by bile ducts or other portal tract elements (Figure 8). Cytoplasm may be normal, clear, and glycogen rich, or fatty. Nuclear atypia and mitoses are generally absent. As discussed, hemorrhage may occur within the HCA nodule itself, or the entire nodule may rupture, leading to subcapsular hematoma and/or hemoperitoneum. Fibrotic changes may occur after hemorrhage [94]. Immunohistochemical and pathologic characteristics of each subtype are summarized in Table 3 [92].

### 4.2. Focal Nodular Hyperplasia

Focal nodular hyperplasia (FNH) is the second most common benign liver tumor in pediatrics, with an estimated incidence of 0.02% in childhood, comprising 2–4% of all pediatric liver tumors [110,111,112]. FNH is more common in adulthood; however, in the pediatric cohort, the median age reported is 8.7 years and is most commonly diagnosed around the age of six to ten, with rare case reports in infants [111,113,114,115]. In addition, there is a female predominance.

#### 4.2.1. Clinical Presentation and Laboratory Findings

Clinically, these patients are usually asymptomatic, and FNH is found incidentally. If symptomatic, the most common presentation is abdominal pain, followed by distension and a palpable abdominal mass. Rupture, hemorrhage, and necrosis is extremely uncommon; however, if large enough, FNH can cause mass effects resulting in portal hypertension and compression of surrounding structures [116]. AFP is classically normal although it has been reported as elevated in cases during infancy, albeit this is more likely secondary to physiologic AFP elevation classic in this age group [111,114]. 

#### 4.2.2. Associated Syndromes and Risk Factors

It is well reported that survivors of childhood cancers—specifically, extrahepatic solid tumors such as Wilms’ tumor and neuroblastoma, as well as hematopoietic stem cell transplant (HSCT) recipients—are at increased risk of development of FNH. It is estimated that the incidence within this specific population is between 5% and 12% [110]. The mean time to develop FNH after treatment has been estimated to be between four and 12 years post therapy, with a shorter interval time period for those individuals who underwent chemotherapy along with HSCT [91]. High doses of alkylating agents such as busulfan or melphalan, liver radiotherapy, and hepatic veno-occlusive disease are reported as risk factors for FNH [117]. The exact pathophysiology has yet to be determined, but FNH is thought to develop as a hyperplastic response of the hepatic parenchyma secondary to circulatory disturbances or via hepatic proliferation induced by vascular injuries such as thrombosis, vascular hyperplasia, or high sinusoidal pressure [116,117]. Cattoni et al. reported a potential association in the development of FNH between iron overload after HSCT, as well as a causal role of estro-progestins in patients receiving hormone replacement therapy after HSCT [110]. 

Children with underlying liver diseases, in particular extrahepatic congenital portosystemic shunts and biliary atresia, are also at high risk of development of FNH. Congenital and surgical portosystemic shunts lead to diversion of intestinal blood to the systemic circulation, bypassing the liver. The precise mechanism is unknown but postulated to be secondary to vascularization of the liver by a large anomalous artery, reactive hyperplasia after hepatocellular injury secondary to vasculitis, or higher blood flow as compared to surrounding areas [91,118]. Patients with biliary atresia, particularly those who have undergone Kasai procedure, are at increased risk of future development of hepatic tumors, with FNH being cited as the most common. These tumors tend to be subcapsular in location, perhaps related to decreased portal flow and/or increased arterial supply, and to lack central scarring [119]. These lesions are proposed to develop as a result of vascular alterations in the setting of chronic hepatic changes [120]. 

#### 4.2.3. Diagnostic Imaging

FNH is quite distinctive radiologically and may be diagnosed solely on imaging findings. Ultrasound is nonspecific but will reveal a homogeneous well-circumscribed lesion which may be isoechoic, hyperechoic, or hypoechoic. Contrast-enhanced ultrasound (CEUS) is an imaging modality that allows detailed characterization of vascular patterns and is mostly useful in lesions greater than three centimeters. Typical findings of FNH on CEUS include sequential centrifugal filling with hyperenhancement on the early portal venous phase and isoenhancement or hyperenhancement on the later portal venous phase. A spoke-wheel appearance of the central arteries and a central scar may also be seen [121]. On CT scan, FNH typically demonstrates uniform enhancement with IV contrast more so than the normal adjacent liver. The central stellate scar, only present in 50% to 70% of cases and more common in larger lesions, will be hyperattenuating on early imaging and enhancing on later images, secondary to retention of contrast material within the myxoid matrix [113,122]. MRI using a hepatocyte-specific contrast agent such as gadoxetate disodium or gadobenate dimeglumine can augment diagnosis. Given that FNH is composed of normal functioning hepatocytes and Kupffer cells, they will demonstrate normal uptake of the hepatocyte-specific contrast agent but then demonstrate abnormal biliary excretion and retention of the agent, appearing isointense to mildly hyperintense as compared to normal surrounding tissue [113,114]. 

Patients with a history of malignancy or underlying liver disease who develop FNH are significantly more likely to have uncommon imaging findings, including multiple lesions that on average are smaller than those in patients without a history of malignancy, as well lesions that demonstrate absence of central scarring [116,119]. These lesions are able to be distinguished from metastatic lesions due to their hyperdense appearance in the arterial or early portal venous phase after administration of IV contrast (as opposed to metastatic lesions, which appear hypodense) [119]. 

##### Histopathology and Immunohistochemistry

Grossly, FNH is typically a well-circumscribed solitary nodular lesion. FNH is composed of a proliferation of bile ducts and a central stellate scar that contains abnormally formed blood vessels (Figure 9). Histologically, FNH arises from polyclonal proliferating cells that are almost identical to surrounding hepatocytes [111,113,114]. Glutamine synthetase, an immunohistochemical stain, is another useful tool in diagnosis. This enzyme is involved in ammonia detoxification and is a downstream target of the Wnt/β-catenin pathway. In a normal liver, expression is limited to hepatocytes that surround the central vein. In FNH, glutamine synthetase is overexpressed resulting in a pathognomonic “map-like” distribution [113]. 

## 5. Malignant Tumors in School-Aged Children and Adolescents

### 5.1. Hepatocellular Carcinoma

Hepatocellular carcinoma (HCC) is the second most common liver cancer in children (31% of all primary hepatic tumors) and constitutes 0.3–0.5% of all pediatric malignancies [123,124]. The incidence is highest within the adolescent population, as opposed to hepatoblastoma, which is primarily seen in children less than five years old. There does not appear to be a racial disparity between African American and Caucasian children with roughly equal incidence reported [123]. The five-year overall survival rate is poor, estimated to be between 13% and 28% [125]. HCC develops in one of two distinct clinical settings: de novo tumors with no evidence of preceding liver damage versus those that develop on a background of underlying liver disease or cirrhosis from metabolic, infectious, and/or vascular causes. A large case series reported by Cowell et al. identified that patients with de novo HCC were more likely to present at an older age, to be at a more advanced stage with metastatic disease, and to have larger tumors than patients with known underlying liver disease [126]. 

#### 5.1.1. Clinical Presentation

Most commonly, HCC presents with an abdominal mass and pain, which may be referred to the back and shoulder, as well as cachexia and jaundice in later stages of disease [127]. The liver on palpation is typically firm and hard and may feel nodular. Patients with underlying cirrhosis can manifest with signs and symptoms of decompensated end-stage liver disease and portal hypertension, such as ascites, splenomegaly, variceal bleeding, spider nevi, digital clubbing, and encephalopathy. Of note, up to one-third of pediatric HCC cases are detected incidentally, and patients are asymptomatic [123,126,127]. Fibrolamellar HCC (FL-HCC) typically arises in patients who do not have underlying liver disease or cirrhosis at the time of tumor development, and patients are classically younger with fewer comorbities [128,129]. In addition, patients commonly present with larger tumor sizes and more advance disease [129]. Clinical presentation is classically nonspecific in FL-HCC with cachexia, abdominal pain, and malaise [129]. A distinguishing clinical feature of FL-HCC is the association of paraneoplastic syndromes, with report of androgen aromatization leading to gynecomastia, as well as with tumoral production of thyroid hormone and beta HCG [130].

#### 5.1.2. Laboratory Findings

AFP, as discussed previously, is an extremely common marker of several hepatic malignancies. Therefore, this marker is useful in surveillance of patients with known cirrhosis or predisposing conditions, as its sensitivity within that group is high, and higher AFP levels at time of diagnosis are seen in patients with higher fatality rates [131]. However, patients with fibrolamellar subtype of HCC, classically seen in de novo HCC, do not typically show elevation in AFP. As a biological marker of hepatobiliary disease and malignancy, albeit not specific to the liver, alkaline phosphatase (ALP) was reported by Cowell et al. to have the highest sensitivity (86%) within their case series [126]. 

#### 5.1.3. Associated Syndromes and Risk Factors

The primary etiology of HCC in endemic areas is hepatitis B (HBV) infection, most frequently acquired congenitally. There is a bimodal age distribution in these patients, with a peak at approximately one year of age and another in adolescence at around 12 to 15 years of age, as well as a male predominance after the age of four [132]. Not surprisingly, with the introduction of mass immunization against hepatitis B in Western countries in the 1980s, the incidence of HBV-related HCC has dramatically declined, with a reported 70% reduction in risk [133]. HBV related HCC is most common in males and is proposed to be related to sex steroids, with increased HBV transcription and suppression of tumor suppression genes by androgens leading to hepatocarcinogenesis [134]. Clinically, patients are more likely to have underlying cirrhosis and portal vein invasion than non-HBV-related HCC, which subsequently leads to a more aggressive and advanced disease [123].

Hereditary tyrosinemia is an autosomal recessive metabolic disorder characterized by lack of fumarylacetoacetate hydrolase, an enzyme that is integral for the breakdown of tyrosine. Loss of this enzyme leads to abnormal accumulation of tyrosine, and its toxic metabolites in the liver lead to hepatotoxicity and increased risk of development of HCC [135]. In addition, these patients may present with renal dysfunction, porphyria-like illness, or cardiomyopathy. Nitisinone, which blocks the enzyme parahydroxyphenylpyruvic acid dioxygenase (involved in the second step of tyrosine degradation) and prevents accumulation of toxic metabolites, is the mainstay of treatment for this disorder. Nitisisone provides protection against HCC if initiated within the first month of life (1% risk); but, if started after the age of two years, a significant risk for HCC remains (25% risk) [123,136]. 

Hepatic fibrosis and HCC are known complications of patients who have undergone Fontan procedures as children, secondary to hepatic congestion and low cardiac output [137,138]. Ten years after Fontan palliation is the expected time point for development of cirrhosis and HCC, although cases have been published with cirrhosis occurring as early as 4–5 years post Fontan [139]. Screening with serial ultrasounds is recommended in this population to aid in early detection of HCC; however, further research in this area is greatly needed as no current consensus guidelines exist. 

Progressive familial intrahepatic cholestasis (PFIC) is a group of disorders known to cause chronic liver disease in children. There are four subtypes, types 1 through 4, each with different genetic mutations. PFIC-2 is characterized by impairment of bile flow secondary to deficiency of the ABCB11 gene, which encodes the bile salt export pump protein leading to constant exposure of hepatocytes to bile salts and results in chronic inflammation with increased risk of carcinogenesis [140]. Notoriously, children with PFIC-2 are at increased risk of development of HCC at an early age (incidence of 15%), although there are rare case reports of HCC developing in types 3 and 4 [126,141]. HCC is often detected incidentally or only picked up at the time of metastatic disease in PFIC-2; therefore, it is recommended that patients are closely monitored with abdominal ultrasound for the potential development of HCC starting at one year of age [141]. 

HCC in the setting of glycogen storage disorders arises from malignant transformation of hepatic adenomas, as opposed to de novo mutations. This occurs most commonly in GSD Type 1a (glucose-6-phosphatase deficiency) in the second to third decade of life [123]. The incidence of malignant transformation is estimated to be between 11% and 14% in various case series [142,143]. Appropriate metabolic control is reported to decrease the incidence of hepatic adenomas but appears to have no effect on occurrence of malignant transformation, and the pathophysiology has yet to be clearly described. In addition, it is reported that AFP levels are frequently normal with development of HCC in this population. Patients with malignant transformation are likely to have a rapid increase in size and/or number of adenomas. Adenomas with hemorrhagic and necrotic changes are more suspicious for HCC, and patients who present with these findings are recommended to undergo contrast ultrasonography as well as biopsy or surgical resection [143].

Biliary atresia, an obstructive cholangiopathy presenting in the neonatal period, is one of the most common causes of chronic liver disease in children. Treatment is mostly surgical, most commonly with the Kasai portoenterostomy. It is estimated that 1% of children with biliary atresia will experience malignant transformation at some point in their lives, occurring as early as infancy. The identification of HCC can be difficult in patients who have undergone the Kasai procedure, as it often causes dominant regenerative areas that may become large and nodular, similar to HCC. As with other chronic liver diseases, these patients are recommended to undergo serial monitoring with AFP levels and ultrasonography to monitor for occurrence of new focal lesions [144].

There are rare case reports of pediatric HCC associated with Budd–Chiari and alpha-1 anti-trypsin disorder, although these much more often develop during adulthood rather than in childhood or adolescence [123,145]. From a vascular-anomalies perspective, congenital extrahepatic portosystemic shunts, which cause shunting of blood between the portal and systemic veins, confer an increased risk of HCC, although the exact etiology is unclear. The majority of case reports are in adults; however, there are reports of development of HCC as young as eight years old [146]. 

FL-HCC has been associated with Byler’s disease, hereditary hemochromatosis, tyrosinemia, PFIC, and alpha-1 antitryspsin deficiency [128].

#### 5.1.4. Diagnostic Imaging

Ultrasound is most commonly the first-line imaging for patients with a suspected abdominal mass and is used for periodic screening in high-risk groups. HCC is classically described as a heterogenous hyperechoic mass with increased vascularity [123]. Tumor detection is particularly difficult in patients with predisposing conditions, as the cirrhotic liver may contain regenerative nodules and dysplastic nodules that are difficult to distinguish from HCC on imaging alone. Contrast-enhanced ultrasound (CEUS) will reveal diffuse or heterogenous arterial enhancement due to peripheral subcapsular vessels, as opposed to rim-like peripheral arterial enhancement, which is classically seen in metastatic lesions [147,148]. Findings of a mass over two centimeters with classic features of HCC on MRI or CT (large, multifocal, hypervascularized with a wash-out phenomenon) has a 95% positive predictive value (Figure 10) [149]. On CT imaging, calcifications are less common than in other hepatic tumors. On contrast-enhanced imaging, HCC shows an intense enhancement in the arterial phase and a contrast wash out in late venous phases. CT is preferred to evaluate the extent of the tumor, resectability, and presence of vascular invasion and/or metastases. 

As discussed earlier, AFP is not typically elevated in FL-HCC; therefore, identification of classical imaging findings becomes paramount. A potentially catastrophic misdiagnosis would be that of FNH as opposed to FL-HCC. On MRI, both are characterized by subtle deviations of signal intensity in comparison to the background liver parenchyma on precontrast T1- and T2-weighted images with arterial hyperenhancement and the presence of a central scar. FL-HCC however, demonstrates greater heterogeneity of texture within the lesion secondary to necrosis and hemorrhage. The central scar is hypointense in T2-weight images, and FL-HCC demonstrates portal venous hypoenhancement. Calcification is more consistent with FL-HCC but may not be reliably identified on MRI [150,151].

#### 5.1.5. Histopathology and Immunohistochemistry

Conventional HCC demonstrates tumor cells that resemble normal hepatocytes with various degrees of cytologic and architectural atypia and are accordingly classified as well-, moderately, or poorly differentiated. The tumor cells contain abundant eosinophilic cytoplasm with nuclei that are centrally located and irregular with abundant mitoses [152]. (Figure 11) These tumor cells are arranged in a trabecular pattern that is similar to normal hepatic architecture with very little intratumoral stroma [153]. 

Those specimens that do not show conventional histology are divided into further subtypes. Given the histopathologic heterogeneity of HCC, only about 35% of all HCCs are able to be further classified into histopathologic variants. These include fibrolamellar, steatohepatic, clear cell, macrotrabecular-massive, scirrhous, chromophobe, neutrophil rich, and lymphocyte rich [153]. 

Fibrolamellar HCC is a rare subtype, accounting for approximately 1% of all HCC, although it is more commonly seen in children and adolescents than in adults [153]. Initially, this histology was thought to be favorable, but it is now accepted that FLL-HCC portends a poor prognosis [154]. It is a well-circumscribed solitary tumor that also often presents with bilobar involvement, as opposed to the multicentric nature of presentation in other subtypes [155]. FL-HCC is distinctive in that it is the only liver tumor that is most commonly seen in the left lobe of the liver [128]. Grossly, it is typically well-circumscribed, hard, scirrhous, and larger than conventional HCC. It may mimic focal nodular hyperplasia, given that it may have a central scarred zone with possible calcification [128,156]. Histologically, FL-HCC has a fibrotic stroma that contains large polygonal cells with a deeply eosinophilic cytoplasm that may contain hyaline globules and pale bodies that are PAS-positive. The name itself is derived from the thick fibrous collagen bands that surround the tumor cells and are characteristic of this subtype [128]. The nuclei are classically large and have prominent eosinophilic nucleoli (Figure 12).

Immunohistochemistry demonstrates reactivity to traditional markers of neoplastic hepatocellular differentiation, including HepPar-1 and glypican-3. FL-HCC demonstrates expression of cytokeratin 7, unlike conventional HCC. Unlike HB, HCC generally does not show nuclear expression of β-catenin [31].

### 5.2. Undifferentiated Embryonal Sarcoma

Undifferentiated embryonal sarcoma of the liver (UESL) is an aggressive tumor of mesenchymal origin that was first described by Stocker and Ishak in 1978 [157]. Classically, it presents in children from the age of six to ten and shows no gender predilection. It is considered to be the third most common malignant hepatic tumor in childhood, after hepatoblastoma and hepatocellular carcinoma, accounting for approximately 9–15% of all pediatric liver tumors, with an incidence of one case per million people per year [30,158,159,160,161,162,163].

#### 5.2.1. Clinical Presentation

The presentation is nonspecific, with abdominal pain, hepatomegaly, nausea, anorexia, and fever, and is rarely associated with jaundice [30,159,160,161,162,163]. The majority of patients who present are asymptomatic; however, patients may present acutely due to tumor rupture and tumor wall dehiscence secondary to rapid growth [163,164,165]. Metastatic disease, commonly to the lungs, peritoneum, and pleura, is reported in approximately 5–13% of patients, although it has been rarely reported to be present at the time of diagnosis [160,162,166].

#### 5.2.2. Laboratory Findings

No distinctive laboratory findings have been reported for UESL. Patients may present with mild leukocytosis or leukopenia, anemia, hypoalbuminemia, mildly elevated transaminases, and inflammatory markers [160,163,167]. AFP and carcinoembryonic antigen are typically normal, although cases with elevated levels have been reported [159,160,161,163,164,165,168]. Interestingly, Letherer et al. reported a case of a UESL in a 15-year-old male that caused a false positive ELISA result for *Echinococcus*, *Entameoba histolytica*, and histoplasmosis secondary to molecular mimicry [168]. Zhang et al. reported a case of a nine-year-old female with elevated IgE and transient peripheral eosinophilia, initially misdiagnosed as *Echinococcus* [165]. Paraneoplastic syndromes and erythropoeitin-secreting capacity have been reported in rare adult cases [169,170].

#### 5.2.3. Associated Syndromes and Risk Factors

To date, there are no known associations between any tumor predisposition syndrome and UESL [30]. Some reported cases of UESL have arisen from malignant transformation of mesenchymal hamartoma of the liver, proposed to be secondary to cytogenetic aberration in the region of chromosome 19q13 [166,171]. Studies have shown that copy number alterations are common; however, none are noted to be specific to UESL. Gains in chromosomes 1q, 5p, and 6q and losses in chromosomes 9p, 11p, and 14 have been reported as recurrent events [30]. Case studies and series have identified mutations in the DNA-binding domain of the TP53 gene [161,172] as well as somatic mutations in WDR25, CMTM1, and DNHA17 [173], although these are reported in single cases and further investigation into the exact pathogenesis is required [30]. Whole exome sequencing was performed in a sample of 14 patients with UESL, which found that the combination of C19MC hyperexpression via chromosomal structural event with TP53 mutation or loss was a highly recurrent genomic feature of UESL [171].

#### 5.2.4. Diagnostic Imaging

Imaging can serve as a valuable diagnostic clue for UESL. UESL will appear as a solid, predominantly hyperechoic mass with focal anechoic areas and cystic portions on ultrasound. CT will reveal a single, well-demarcated, predominantly hypoattenuated cystic mass with internal septations. A crucial diagnostic clue for UESL is the discrepancy between the solid appearance on ultrasound and the cystic appearance on CT imaging, likely secondary to the high water content of the prominent myxoid stroma. In addition, the presence of serpiginous vessels within the tumor on CT imaging is proposed to be unique to UESL. MRI, which is the preferred imaging modality for preoperative planning due to its ability to detect vascular invasion, biliary obstruction, and hilar adneopathy, will demonstrate a well-demarcated mass with T2 hyperintensity and T1 hypointensity [30,160,162,163,164,166].

#### 5.2.5. Histopathology and Immunohistochemistry

Diagnosis relies on postoperative pathologic examination. Grossly, the lesion is typically a large (≥10 cm), spherical, solitary lesion that most often arises from the right lobe of the liver [30,162]. UESL often has clear tumor borders with a fibrous pseudo capsule compressing adjacent parenchyma. The cut surface is yellow to tan with a heterogenous appearance that is predominantly solid with alternating cystic foci and areas of gelatinous degradation (corresponding to areas of myxoid histology). Necrotic and hemorrhagic areas are common [30,160,163].

Histologically, UESL will have hypercellular sheets of highly pleomorphic tumor cells with a myxoid background (Figure 13). This tumor demonstrates a high mitotic index, frequent atypical mitoses, and apoptotic bodies and a high Ki67 proliferation index (30%) [30,161]. Cytology is notable for medium to large spindle or stellate-shaped cells with ill-defined borders and inconspicuous nucleoli within a myxoid matrix or fibrous stroma [30,160,164]. Multinucleated giant cells, bizarre cells with aberrant nuclei, and eosinophilic hyaline globules (which are PAS-positive) may also be identified, which are important diagnostic clues [30,160,163,164,165]. At the periphery of the tumor, trapped hepatocytes and bile duct cells may be visible [160,165].

Immunohistochemical stains are nonspecific and nondiagnostic for UESL, although they can aid in exclusion of other tumors within the differential [30,160,163,165].

## 6. Conclusions

The differential diagnosis for intra-abdominal and liver masses in the pediatric population is broad. Familiarity with characteristics unique to each specific tumor as well as potential clinical complications will aid the clinician in raising the index of suspicion and guide selection of appropriate imaging and laboratory evaluations prior to definitive histopathologic diagnosis. As described above, defining characteristics unique to specific tumors have been recognized and delineated for a number of pediatric hepatic tumors. However, in less common tumors, the majority of evidence is based upon case series and case reports, and further research is needed in these areas in order to provide guidance on appropriate diagnostic tools and management strategies. We believe that this review article provides fundamental information for the clinician regarding age-specific differential diagnoses, appropriate imaging, and laboratory tests, and unique histopathological findings in liver tumors in children. Ultimately, utilization of the appropriate diagnostic modalities together with a multidisciplinary approach of specialists will ensure accurate diagnosis and proper treatment of these tumors. 

## Figures and Tables

**Figure 1 diagnostics-11-00333-f001:**
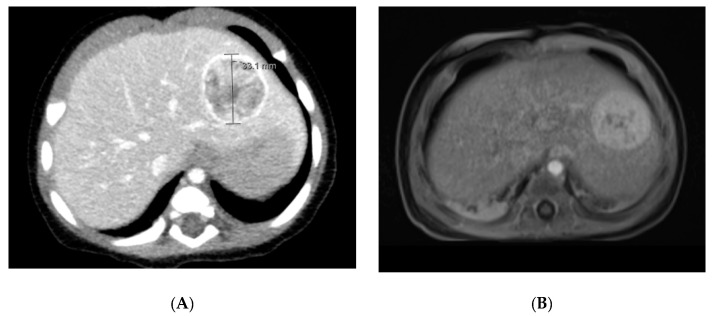
CT and MRI Imaging of Unifocal Hemangioma. (**A**) CT: The liver is remarkable for a well-circumscribed rounded lesion in the left lobe of the liver. This measures approximately 3.3 cm in anterior–posterior dimension by 3.1 cm in transverse dimension by 2.4 cm in craniocaudal dimension. There is intense peripheral enhancement with mixed enhancement of the internal portion of this lesion. (**B**) MRI: A large T2 hyperintense well circumscribed mass is seen in left lobe within segment 2 and 3 of the liver measuring 2.9 × 2.9 × 2.5 cm in anterior-posterior, transverse and craniocaudal dimensions. It demonstrates peripheral nodular enhancement with centripetal contrast filling.

**Figure 2 diagnostics-11-00333-f002:**
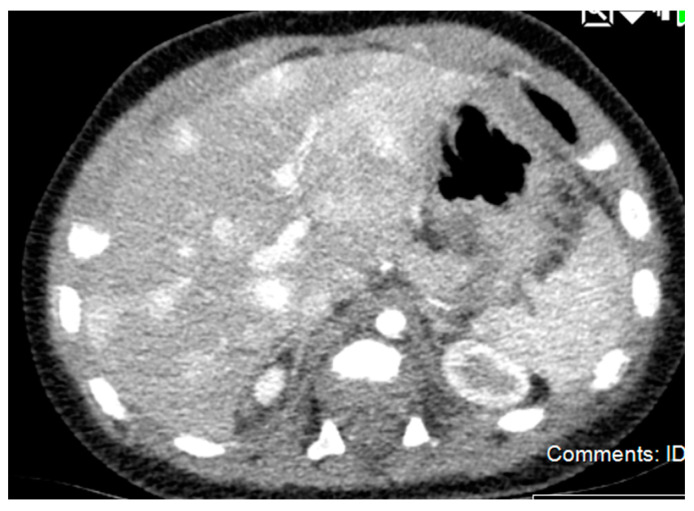
CT Imaging of Multifocal Hemangioma. There are numerous foci of intense enhancement on early postcontrast imaging throughout the liver, which become isointense to surrounding liver parenchyma on delayed imaging.

**Figure 3 diagnostics-11-00333-f003:**
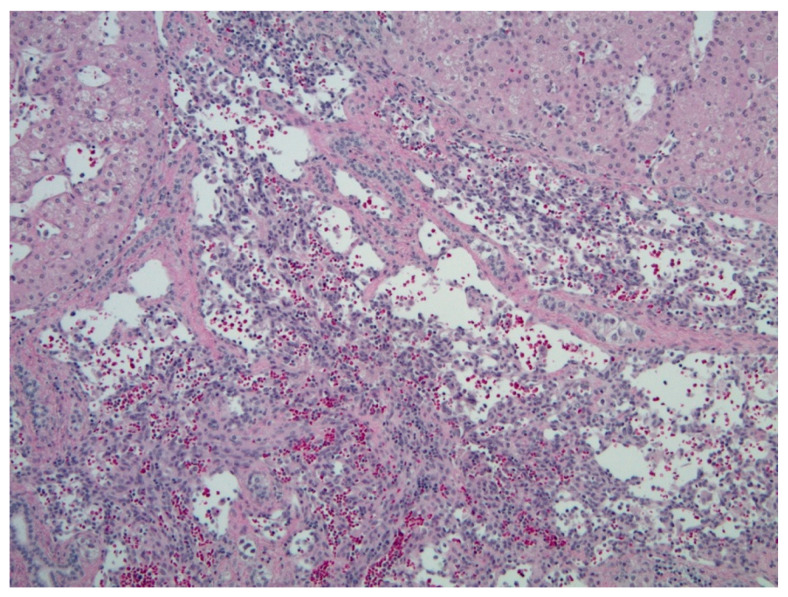
Hepatic hemangioma (hematoxylin & eosin (H&E), 10× magnification). Vascular channels lined by benign-appearing endothelial cells, with surrounding hepatic parenchyma.

**Figure 4 diagnostics-11-00333-f004:**
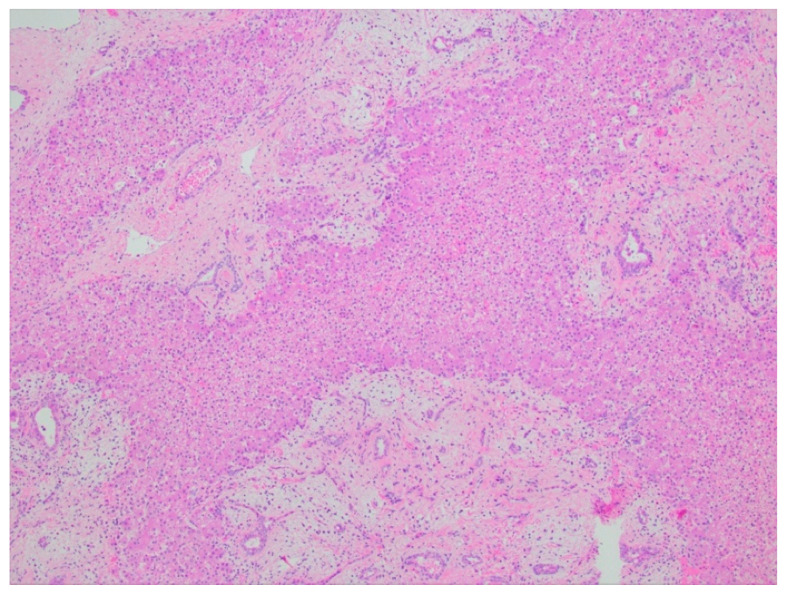
Mesenchymal hamartoma (H&E, 4× magnification). Mixture of epithelial and stromal elements, with interspersed hepatic parenchyma. The epithelial component consists of disorganized bland ductal structures, and the stromal component includes spindle cells, set in a loose myxoid background.

**Figure 5 diagnostics-11-00333-f005:**
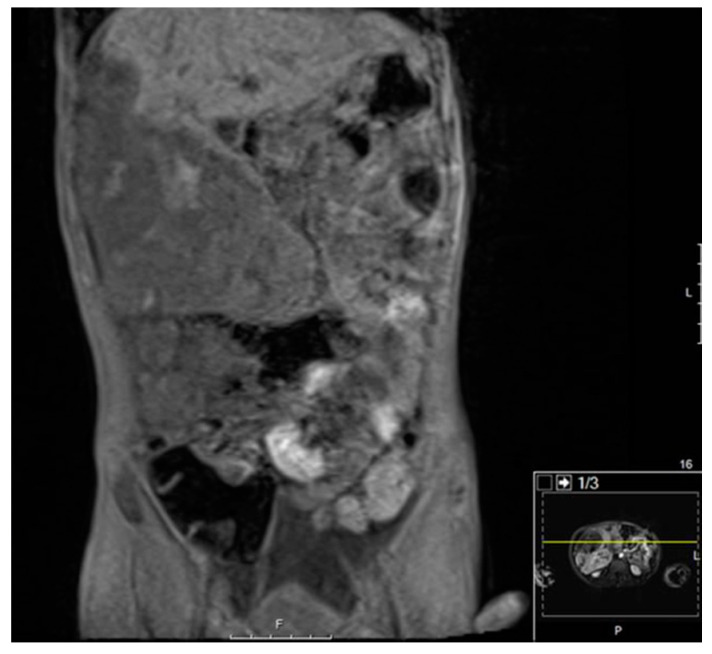
CT Imaging of Hepatoblastoma. There is a lobulated heterogeneous mass involving the fifth and sixth segments of the right lobe, which also contains well-defined rounded hypoattenuating areas. No calcification can be seen. The mass extends inferiorly from the right lobe of the liver as well as extending lateral and anterior to the liver at its superior aspect. The mass measures approximately 21.7 cm in greatest sagittal dimension and 10.6 cm in greatest transverse dimension and 10.7 cm in greatest AP dimension. Superiorly, the mass appears to displace the anterolateral aspect of the liver posterior and medially and inferiorly, the mass displaces bowel posteriorly and medially.

**Figure 6 diagnostics-11-00333-f006:**
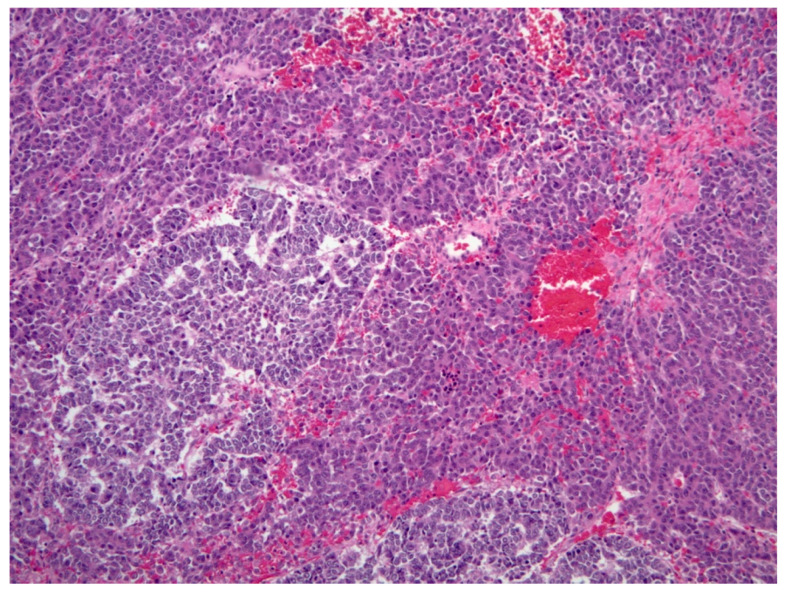
Hepatoblastoma (H&E, 10× magnification). Mixed embryonal (left) and fetal hepatoblastoma. The embryonal component demonstrates a higher nuclear-to-cytoplasmic (N:C) ratio and forms rosette-like structures, while the fetal component recapitulates the appearance of fetal hepatocytes.

**Figure 7 diagnostics-11-00333-f007:**
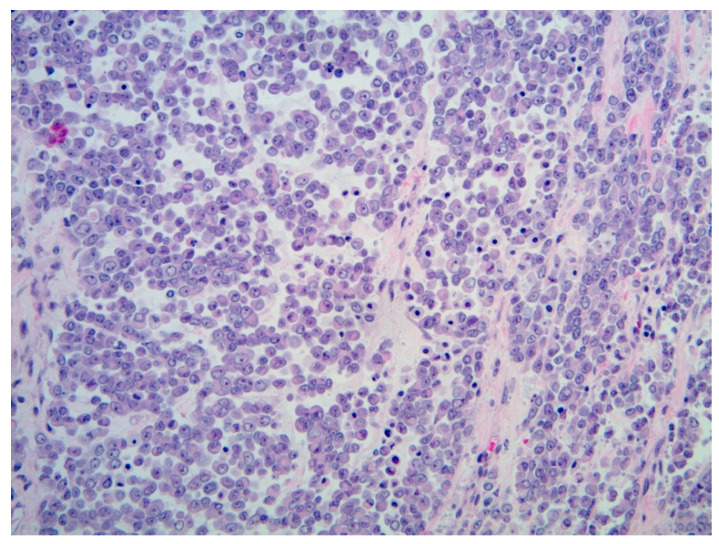
Malignant rhabdoid tumor (H&E, 20× magnification). Typical rhabdoid cells, with eccentric nuclei with prominent nucleoli, and eosinophilic cytoplasm, with discohesive architecture.

**Figure 8 diagnostics-11-00333-f008:**
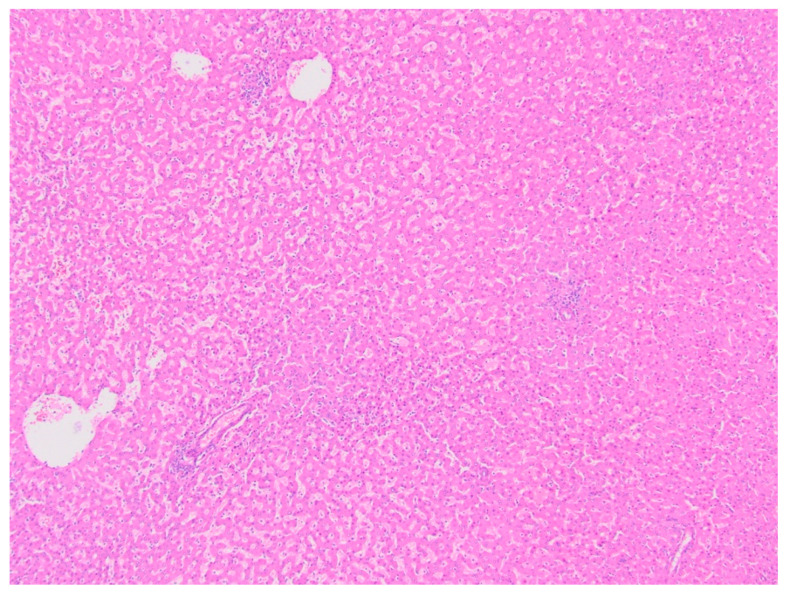
Hepatic adenoma (H&E, 4× magnification). Hepatic parenchyma with no cytologic atypia and effacement of normal lobular architecture, with lack of portal tracts.

**Figure 9 diagnostics-11-00333-f009:**
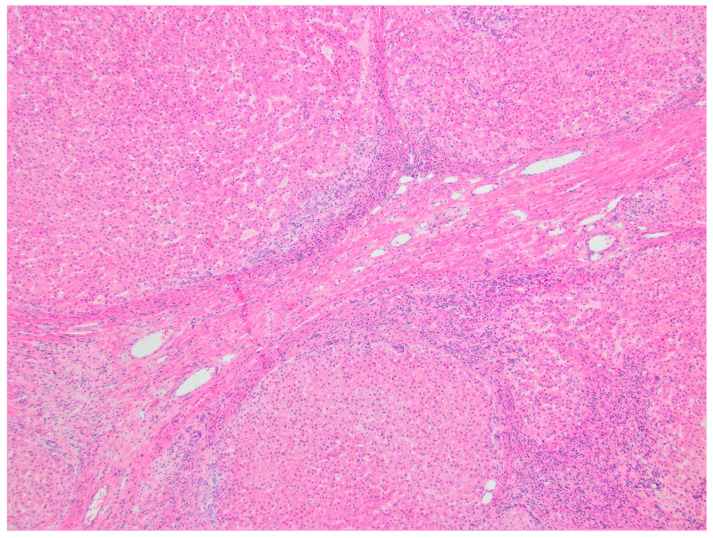
Focal nodular hyperplasia (H&E, 4× magnification). Nodular hepatic parenchyma with bile ductular proliferation and central scar containing abnormal blood vessels.

**Figure 10 diagnostics-11-00333-f010:**
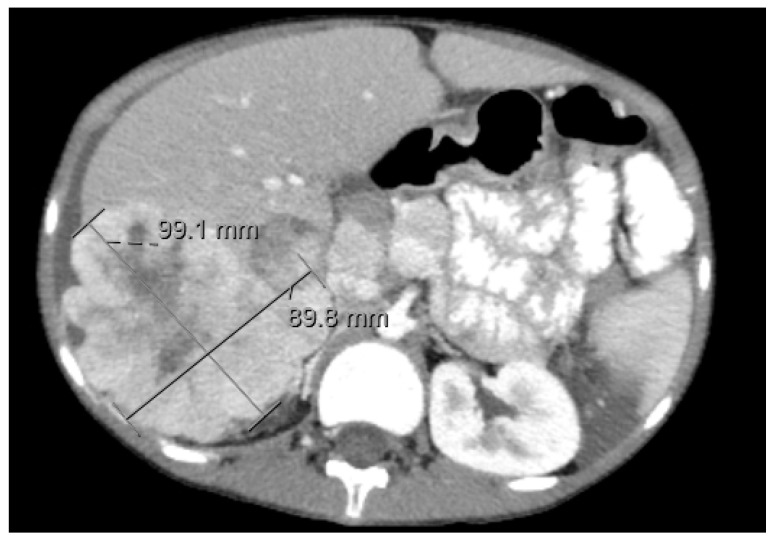
CT Imaging of Hepatocellular Carcinoma. Irregular heterogeneously enhancing mass is present in the right hepatic lobe, extending exophytically, laterally, and inferiorly from the liver surface, measuring up to 9.9 × 9.0 cm in maximum transaxial diameters with central hypoattenuation, suggesting necrosis. There is nodular extension along the peritoneal surface lateral to the right hepatic lobe, as well as more anteriorly.

**Figure 11 diagnostics-11-00333-f011:**
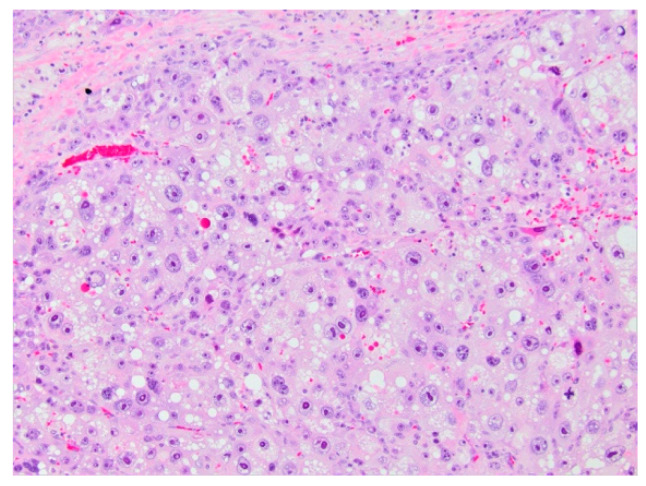
Conventional hepatocellular carcinoma (H&E, 10× magnification). Malignant proliferation of hepatocytes, with pleomorphism, large, prominent nucleoli, and abundant mitotic figures, including atypical forms.

**Figure 12 diagnostics-11-00333-f012:**
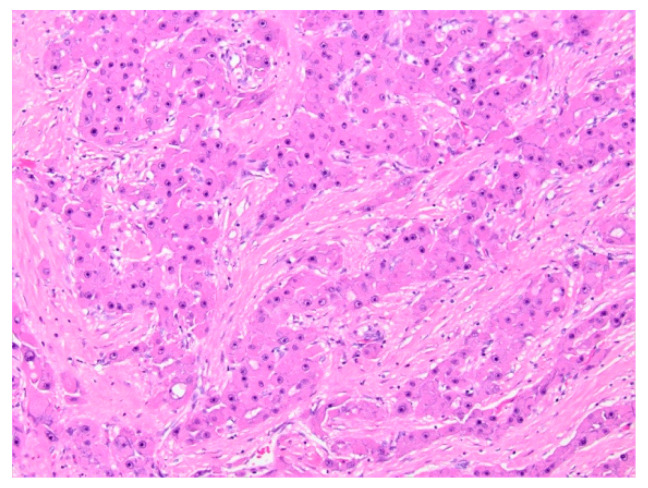
Fibrolamellar type hepatocellular carcinoma (H&E, 10×). Malignant proliferation of hepatocytes with abundant, granular eosinophilic cytoplasm with prominent nucleoli, set in a fibrotic stroma.

**Figure 13 diagnostics-11-00333-f013:**
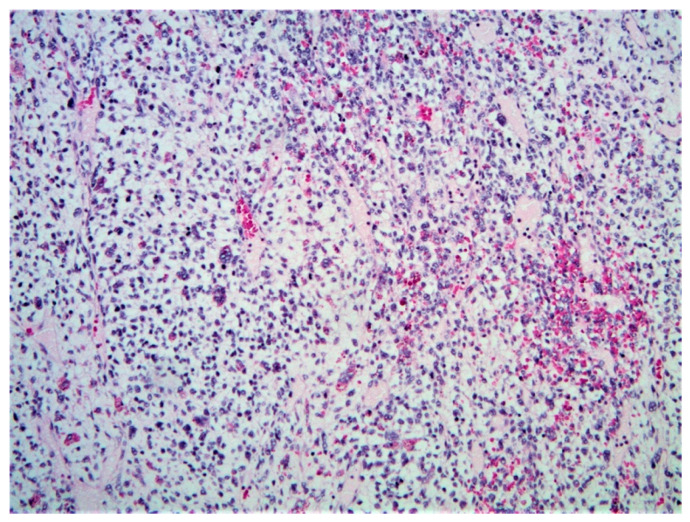
Undifferentiated embryonal sarcoma (H&E, 10× magnification). Sheets of highly pleomorphic cells with abundant mitotic figures, set in a myxoid stroma.

**Table 1 diagnostics-11-00333-t001:** Infantile Hepatic Hemangiomas—Clinical Presentation.

	Focal Hepatic Hemangiomas (FHH)	Multifocal Hepatic Hemangiomas (MHH)	Diffuse Hepatic Hemangiomas (DHH)
**Age of Onset**	Proliferates in utero Fully formed at birth	Post-natal period	Post-natal period
**Natural Course of Progression**	Undergoes involution over the first 12–14 months of life	3 Phases: (1) Proliferation—peaks around 6 months of age (2) Early involution—around 10 months of age (3) Late involution	Significant liver involvement with near complete displacement of all liver parenchyma
**Laboratory Findings**	Anemia Mild thrombocytopenia Elevated alpha feto-protein (AFP) at birth (should downtrend) Normal thyroid studies	Low T4 High TSH	Low T4 High TSH
**Clinical Presentation**	Asymptomatic prenatally After birth may present with abdominal distention and palpable hepatic mass, unexplained anemia, or coagulopathy No association with cutaneous infantile hemangiomas	Most asymptomatic May present with hepatomegaly and abdominal distension or be identified on screening after development of congestive heart failure or hypothyroidism May be associated with cutaneous infantile hemangiomas (60%)	Massive hepatomegaly with hepatic failure May be associated with cutaneous infantile hemangiomas
**Clinical Complications**	Prenatally: fetal cardiomegaly, cardiac failure, hydrops fetalis Postnatally: cardiac failure	High output cardiac failure	High output cardiac failure Abdominal compartment syndrome Multisystem organ failure Profound consumptive hypothyroidism

**Table 2 diagnostics-11-00333-t002:** Histopathology of Hepatoblastoma.

Subtype	Cell Size, Shape, and Pattern	Nuclei and Nucleoli	Cytoplasm
**Fetal**	Polygonal cells between 10 and 20 μ in diameter Organized in sheets or as one to two cell thick trabecula	Centrally placed round nuclei with well delineated nuclear membranes, finely stippled chromatin and inconspicuous nucleoli	Appears clear, commonly contains clusters of hematopoietic precursors Crowded pattern may have ampophilic cytoplasm with a proportionately high nuclear-to-cytoplasmic ratio
**Embryonal**	Round or angulated cells, 10 to 15 μ in diameter Grow in sheets and commonly form tubular or acinar structures around a central lumen	High nuclear-to-cytoplasmic ratio	Scant cytoplasm
**Small Cell**	Round or oval, 7–8 μ in diameter Usually form clusters or nests in an “organoid” pattern	Fine nuclear chromatin, inconspicuous nucleoli, minimal mitotic activity	Scant cytoplasm
**Macrotrabecular**	Cell plates more than 20-cells thick that can be found in a pure form or in combination with other patterns	Variable (may be reflective of fetal or embryonal origins or be pleomorphic)	Variable

**Table 3 diagnostics-11-00333-t003:** Hepatocellular Adenomas (HCA): Histology, Immunohistochemistry, and Genetic Mutations.

	Histology	IHC	Genetics
**HHCA**	Intralesion steatosis with a lack of associated inflammation or cellular atypia	Decreased or absent LFABP immunostaining	HNF1A inactivating mutation
**IHCA**	Inflammatory infiltrate, sinusoidal dilatation, dystrophic arteries, and variable ductular reaction in periphery of the lesions	Positive CRP Positive serum amyloid A	IL6ST mutation
**bHCA**	Mild cytologic atypia Pseudoacinar formation	Diffuse and strong expression of glutamine synthetase Nuclear positivity for β-catenin	CTNNB1 mutation
**shHCA**	Intratumoral hemorrhage	Positive Prostaglandin D2 synthase	Deletion of INHBE leading to INHBE–GLI1 fusions
**uHCA**	Nonspecific, has typical HCA findings	None	None

HHCA: *HNF1A*-inactivated hepatocellular adenoma; IHCA: Inflammatory hepatocellular adenoma; bHCA: *Beta-catenin*-mutated hepatocellular adenoma, shHCA: Sonic hedgehog-activated hepatocellular adenoma; UHCA: Unclassified hepatocellular adenoma; IHC: immunohistochemistry; LFABP: Liver fatty acid binding protein; CRP: C-reactive protein.

## Data Availability

Not applicable.

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
