# Peer review of "Pediatric Primary Hepatic Tumors: Diagnostic Considerations"

_diagnostics, 2021, doi:10.3390/diagnostics11020333_

Round 1

Reviewer 1 Report

Overall very nice review of liver tumors in pediatric patients. Minor point is that IHC isn't necessarily crucial in the diagnosis of hepatoblastoma - this is overstated a bit. But all in all, this is a good review.

Author Response

Thank you for your review. We have edited the language to reflect your comment (Lines 355/356).

Reviewer 2 Report

I read with interest this fascinating review. There are a few aspects that need to be fixed before going to be published. The authors indicate that RICH and NICH are difficult to distinguish on imaging, but serial imaging may show a subsequent decrease in the size of the lesion suggestive of RICH as opposed to NICH. There are also specific criteria in imaging that need to be included in the review other than ref. 4. Moreover, the manuscript is quite dry. We need 8-10 radiologic imaging or histopathology microphotographs. The section of angiosarcoma needs a table including all cases of angiosarcoma in children. This is a comprehensive review and we would like it is cited properly. Is ductal plate malformation a risk factor for BA developing HCC? If it is not, why? Please discuss! The fibrolamellar HCC needs to be expanded. Please cite and discuss this tumor properly: Sergi CM. Hepatocellular Carcinoma, Fibrolamellar Variant: Diagnostic Pathologic Criteria and Molecular Pathology Update. A Primer. Diagnostics (Basel). 2015 Dec 30;6(1):3. doi: 10.3390/diagnostics6010003. PMID: 26838800; PMCID: PMC4808818. 

Author Response

The authors indicate that RICH and NICH are difficult to distinguish on imaging, but serial imaging may show a subsequent decrease in the size of the lesion suggestive of RICH as opposed to NICH. There are also specific criteria in imaging that need to be included in the review.

  • Clarified the wording of this (Lines 91-115) and expanded imaging criteria as well as citations. 

We need 8-10 radiologic imaging or histopathology microphotographs. 

  • Figures have been added throughout the text

The section of angiosarcoma needs a table including all cases of angiosarcoma in children. 

  • Table is uploaded as a separate document along with the manuscript - it is quite lengthy however so it is difficult to fit within the body of the text - perhaps someone will be able to help with formatting?

Is ductal plate malformation a risk factor for BA developing HCC? If it is not, why? Please discuss! 

  • We were unable to find any direct studies linking HCC with patients with BA attributed to ductal plate malformation. 

The fibrolamellar HCC needs to be expanded. 

  • HCC section re-written to elaborate more on HCC. Thank you for the reference! (Lines 842-860)